# Optimization of Shape Design of Grommet through Analysis of Physical Properties of EPDM Materials

**Young Shin Kim [1], Eui Seob Hwang [2] and Euy Sik Jeon [3,\*]**

[1]  Industrial Technology Research Institute, Kongju National University, Kongju 31080, Korea;
    people9318@gmail.com
[2]  Company-affiliated R&D center/New Business Manager, DAESUNG HI-TECH Co., Ltd., Cheonan 31025,
    Korea; hus5174@dscmc.com
[3]  Department of Mechanical Engineering, Industrial Technology Research Institute,
    Kongju National University, Kongju 31080, Korea
\*  Correspondence: osjun@kongju.ac.kr; Tel.: +82-41-521-9284

**Abstract:** Ethylene propylene diene monomer (EPDM) has superior mechanical properties, water resistance, heat resistance, and ozone resistance. It can be applied to various products owing to its low hardness and high slip resistance properties. A grommet is one of the various products made using EPDM rubber. It is a main component of automobiles, in which it protects wires throughout the inside and outside of a vehicle body. The grommet, made of EPDM, has different mounting performance depending on the process parameters and the shape of the grommet. This study conducted optimization to improve the mounting performance of a grommet using EPDM materials. The physical properties of the main molding materials were investigated according to process parameters. A grommet was fabricated according to the process parameters of fabrication. Insertion force and separation force were examined through experiments. Nonlinear material constants were determined through uniaxial and biaxial tensile tests. The nonlinear analysis of the grommet was conducted, and a compound design that incorporated the shape parameters for the minimum load of each part was derived. Then, additional nonlinear analysis was performed. This was followed by a comparative analysis of the actual model through experimental evaluation.

**Keywords:** Ethylene-propylene diene monomer rubber EPDM; grommet; physical properties; optimization of shape design

---

## 1. Introduction

Ethylene propylene diene monomer (EPDM) rubber is a terpolymer in which ethylene, propylene, and diene are irregularly bonded. Compared to general rubbers, it has superior mechanical properties, water resistance, heat resistance, and ozone resistance. In addition, it has high inheritance and corona discharge resistance because of limited in its chemical structure [1]. Moreover, it can be applied to various products owing to its low hardness and high slip resistance properties [2,3]. Numerous studies have been conducted based on the diverse applications of EPDM, with a focus on the reliability of EPDM-based products [4–6]. Studies have been conducted to analyze the physical characteristics of composite materials [7,8].

A grommet is one of the various products made using EPDM rubber [9]. It is a main component of automobiles, in which it protects wires throughout the inside and outside of a vehicle body. Unlike conventional plastic, EPDM rubber is characterized by high flexibility, high elasticity, and high tensile strength. It is fabricated through an injection molding process. Grommets are also produced through the injection molding process of EPDM [7]. The parameters of this process, such as time and

temperature, not only change the physical properties of raw materials but also affect the insertion force and separation force generated during mounting a grommet on a body when molding a grommet product. Accordingly, in this study, to analyze the physical properties of raw materials according to process parameters, we set the main factors of molding process parameters using a design of an experimental method (DOE) [10–12]. Specimens were prepared according to process conditions. Tensile strength and elongation were measured, and the correlation was analyzed.

The process parameters of EPDM raw materials were set, and the experimental design was established by applying factorial designs from among experimental design methods. The physical properties of the raw materials were tested using the standard test method for soft vulcanized rubber (KS M 6518) [13] for confirming the changes in the physical properties according to process parameters. The physical properties of EPDM were checked and reflected during grommet vulcanization [14]. The experimental design was established using factorial designs among the bellows type, cable type, and cable-less type. Then, we compared and analyzed the maximum insertion force and maximum separation force generated during mounting. It was confirmed that an insertion force and separation force tended to not occur depending on process parameters. We conducted the nonlinear analysis of EPDM to improve grommet design for mounting performance. For this purpose, stress–strain rate information was obtained through uniaxial and biaxial tensile tests, and the nonlinear material constants required for the analysis were determined. The shape parameters of the grommet were set, and a mounting performance simulation was conducted for various shapes. Based on the analytical results, we derived the dimensions for optimizing grommet mounting performance. Additional analysis was conducted with the derived dimensions. An actual grommet was manufactured and analyzed to verify its feasibility.

## 2. Experimental Analysis Based on Molding Process Conditions

### 2.1. Analysis of Physical Properties Based on Molding Process Conditions

The physical properties of EPDM rubber, which is a raw material for making grommet, were analyzed according to the conditions of the injection process. The controllable factors that were expected to affect the physical properties were set, and specimens were fabricated according to the KS M 6518 [13] standard for each experimental condition. The physical properties were set as tensile strength and elongation, which were measured using a universal testing machine. Table 1 shows the main process parameters for injection molding. The experiment was conducted 16 times. The table shows the experimental conditions for each process value.

**Table 1.** Experimental conditions for different process values (DOE).

| No. | Temp. (°C) | Time (s) | Degassing | Strength (Mpa) | Elongation (%) | No | Temp (°C) | Time (s) | Degassing | Strength (Mpa) | Elongation (%) |
|---|---|---|---|---|---|---|---|---|---|---|---|
| 1 | 160 | 200 | ○ | 11.2 | 969.0 | 9 | 160 | 200 | X | 10.5 | 968.3 |
| 2 | 160 | 200 | ○ | 11.1 | 960.4 | 10 | 160 | 200 | X | 10.8 | 944.8 |
| 3 | 160 | 600 | ○ | 13.4 | 853.9 | 11 | 160 | 600 | X | 13.5 | 851.5 |
| 4 | 160 | 600 | ○ | 12.9 | 815.2 | 12 | 160 | 600 | X | 13.0 | 821.2 |
| 5 | 180 | 200 | ○ | 13.2 | 840.5 | 13 | 180 | 200 | X | 12.8 | 838.7 |
| 6 | 180 | 200 | ○ | 13.1 | 836.0 | 14 | 180 | 200 | X | 13.2 | 844.8 |
| 7 | 180 | 600 | ○ | 13.4 | 775.0 | 15 | 180 | 600 | X | 13.0 | 749.7 |
| 8 | 180 | 600 | ○ | 13.7 | 792.3 | 16 | 180 | 600 | X | 13.3 | 767.1 |

Figure 1 shows the main effects and interactions of the factors that affect tensile strength. Temperature and time affect tensile strength, while degassing does not. Additionally, interactions occur according to time and temperature and no interactions occur, owing to degassing.

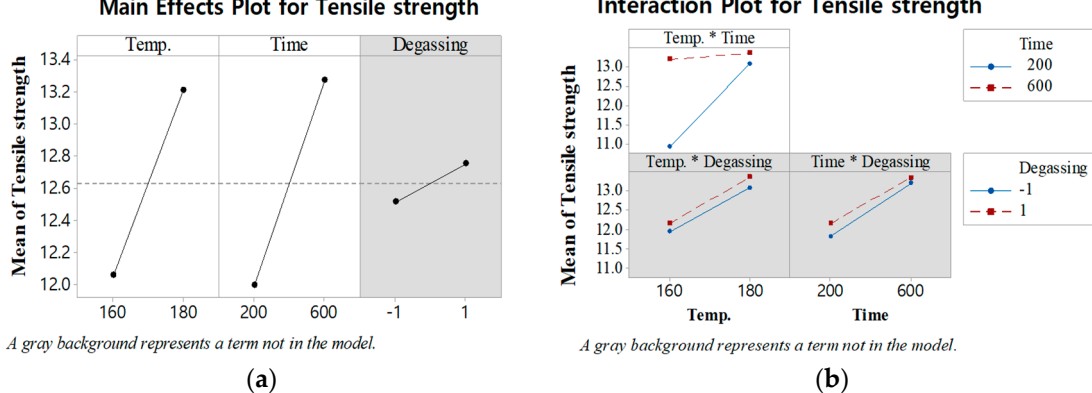

**Figure 1.** Main effects plot and interaction plot for tensile strength. (**a**) Main effects plot for tensile strength; and (**b**) Interaction plot for tensile strength.

Table 2 shows the analysis of variance (ANOVA) result for the factors that affect tensile strength. ANOVA is a collection of statistical models and their associated estimation procedures (such as the "variation" among and between groups) used to analyze the differences among group means in a sample. 'Adj. SS' represents the sum of squares, 'Adj. MS' is the mean squares, and '*F*-Value' is the value of the adj. The SS of each factor divided by the mean squares error. '*p*-value' was derived based on *F* value. *p* values larger than 0.05 are pooled as error terms, and only significant factors are shown. The regression Equation (1) was derived through ANOVA analysis. The $R^2$ value of the regression equation is 94.59% and the adj. $R^2$ value is 93.24%.

$$tensile\ strenght\ =\ -15.59 + 0.1585Temp + 0.04592Time - 0.00025Temp \times Time \tag{1}$$

**Table 2.** ANOVA result for tensile strength.

| Source | DF | Adj SS | Adj MS | *F*-Value | *p*-Value |
|---|---|---|---|---|---|
| Model | 3 | 15.95 | 5.31661 | 69.95 | 0.000 |
| Linear | 2 | 11.91 | 5.95374 | 78.33 | 0.000 |
| Temp (°C) | 1 | 5.380 | 5.38007 | 70.78 | 0.000 |
| Time (s) | 1 | 6.5274 | 6.52741 | 85.88 | 0.000 |
| 2-way interactions | 1 | 4.0423 | 4.04234 | 53.18 | 0.000 |
| T (°C) × Time (s) | 1 | 4.0423 | 4.04234 | 53.18 | 0.000 |
| Error | 12 | 0.9121 | 0.0760 | | |
| Total | 15 | 16.8619 | | | |

### 2.2. Elongation according to Time and Gas Removal Conditions

Figure 2 shows the plot of the major factors that determine elongation. Time and temperature affect elongation, and gas removal has a minor effect on elongation. Figure 2b shows the diagram of the interactions. It can be confirmed that temperature and time affect each other.

Table 3 shows the ANOVA results for the analysis of the factors that affect elongation. Based on the analysis of tensile strength according to process parameters, temperature and time affect tensile strength and elongation, while degassing does not. However, a few specimens without degassing did exhibit pores. We set the degassing parameter in additional experiments. The regression Equation (2) was derived through ANOVA analysis. The $R^2$ value of the regression equation is 96.69% and the adj. $R^2$ value is 95.86%.

$$elongation\ =\ 2213 - 7.436Temp - 1.437Time - 0.00702Temp \times Time \tag{2}$$

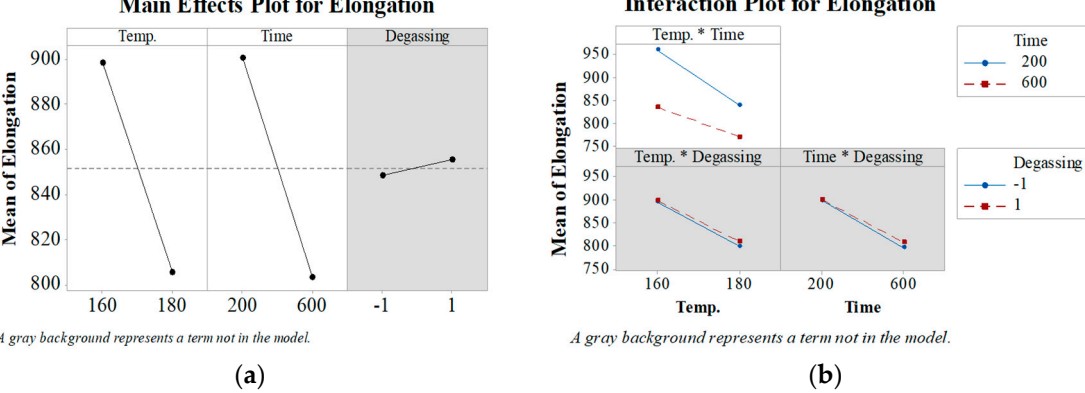

**Figure 2.** Main effects plot and interaction plot for elongation. (**a**) Main effects plot for elongation; and (**b**) Interaction plot for elongation.

**Table 3.** ANOVA result for elongation.

| Source | DF | Adj SS | Adj MS | *F*-Value | *p*-Value |
|---|---|---|---|---|---|
| Model | 3 | 75,096.2 | 25,032.1 | 116.87 | 0.000 |
| Linear | 2 | 71,937.7 | 35,968.9 | 167.93 | 0.000 |
| Temp. (°C) | 1 | 34,243.5 | 34,243.5 | 159.88 | 0.000 |
| Time (s) | 1 | 37,694.2 | 37,694.2 | 175.99 | 0.000 |
| 2-way interactions | 1 | 3158.4 | 3158.4 | 14.75 | 0.034 |
| T (°C) × Time (s) | 1 | 3158.4 | 3158.4 | 14.75 | 0.002 |
| Error | 12 | 2570.2 | 214.2 | | |
| Total | 15 | 77,666.4 | | | |

### 2.3. Measurement of Grommet Mounting Performance according to Molding Process Parameters

We derived the process parameters that increase mounting performance by measuring mounting performance according to grommet shape. Based on the results described in the previous section, temperature and time were set as the process parameters because they affect tensile strength and elongation. We set the maximum and minimum values for each factor according to grommet shape and fabricated the grommet. Here, degassing was applied in the fabrication of all products. The mounting performance of the products was analyzed by measuring the insertion force required for fastening the grommet and the required separation force. Insertion force and separation force were measured using a universal tensile tester when the grommet was inserted into or removed from a panel fixing jig. The experimental speed was set as 50 mm/min, and the experiment was repeated twice. We employed three widely used types of shapes for the grommet. The temperature and time parameters for molding the grommet of each shape in the initial test are shown in Table 4. Figure 3 shows the types of grommet shape.

**Table 4.** Experimental conditions.

| Type | Temp. (°C) | | Time (s) | |
|---|---|---|---|---|
| | Min(−1) | Max(1) | Min(−1) | Max(1) |
| Bellows | 160 | 180 | 200 | 600 |
| Cable | 170 | 190 | 400 | 800 |
| Blank | 180 | 200 | 300 | 900 |

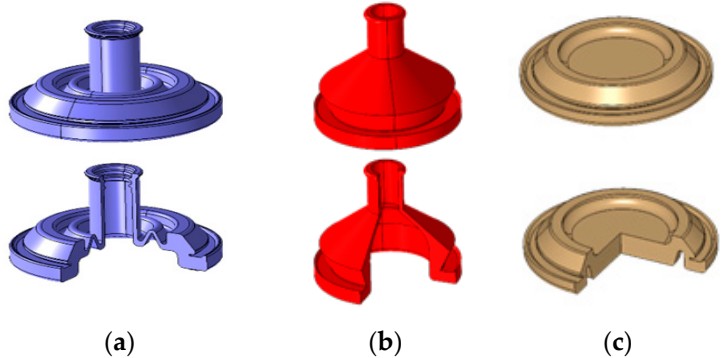

**Figure 3.** Grommet shapes. (**a**) Bellows type; (**b**) Cable type; and (**c**) Cable-less type.

The insertion force and separation force for different grommet shape types are given in Table 5. ANOVA was used to analyze the insertion force and separation force for each type of shape, and it was confirmed that there was no difference between insertion force and separation force according to grommet process parameters. As shown in Figure 4, even though there is no difference between insertion force and separation force according to process parameters, the times at which the maximum insertion force and maximum separation force occur vary depending on process parameters. This appears to be because the elongation rate changes according to process parameters. Moreover, it was confirmed that the change in insertion force and separation force was more influenced by the changes in the shape of the grommet.

**Table 5.** Experimental results.

| Process Value | | Bellows Type | | Cable Type | | Cable-Less Type | |
|---|---|---|---|---|---|---|---|
| Temp. | Time | Insertion Force | Separation Force | Insertion Force | Separation Force | Insertion Force | Separation Force |
| −1 | −1 | 94.1 | 85.3 | 226.5 | 120.6 | 99.0 | 73.5 |
| 1 | 1 | 94.1 | 88.3 | 268.7 | 119.6 | 97.1 | 65.7 |

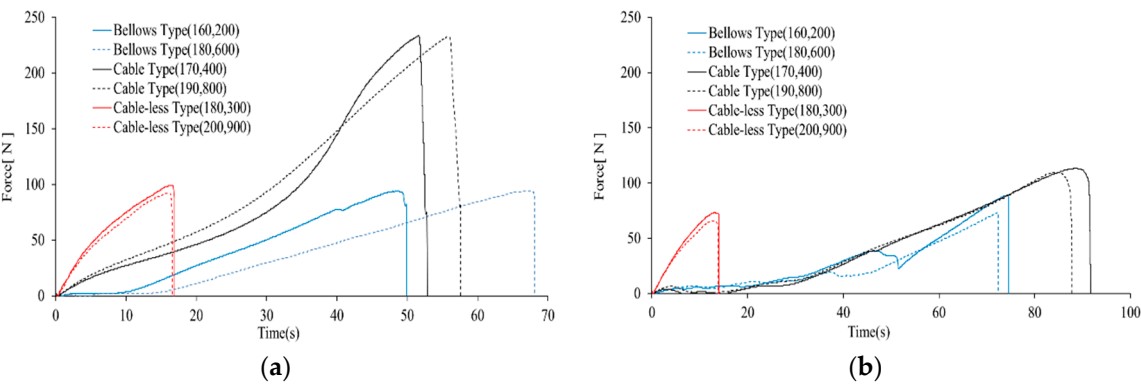

(**a**)  (**b**)

**Figure 4.** Insertion force and separation force experimental results according to process parameters by grommet type. (**a**) Insertion force results according to grommet shape type; and (**b**) separation force results according to grommet shape type.

## 3. Nonlinear Analysis Using FEM

### 3.1. Parameter Settings According to Shape

This study considered that the factors that influenced insertion force and separation force were more affected by the shape of the product than by the process parameters of the product. Therefore, the shape of the product was parameterized to analyze insertion force and separation force according

to the changes in shape. Figure 5 shows the shape parameters of the main part of the grommet, and Table 6 shows the values of each shape parameter.

**Table 6.** Parameters according to shape.

| Level | a (mm) | b (mm) | c (mm) | d (mm) | e (°C) |
|-------|--------|--------|--------|--------|--------|
| 1 | 4 | 10 | 1 | 5 | 135 |
| 2 | 4.5 | 11 | 1.5 | 6 | 145 |
| 3 | 5 | 12 | 2 | 7 | 155 |

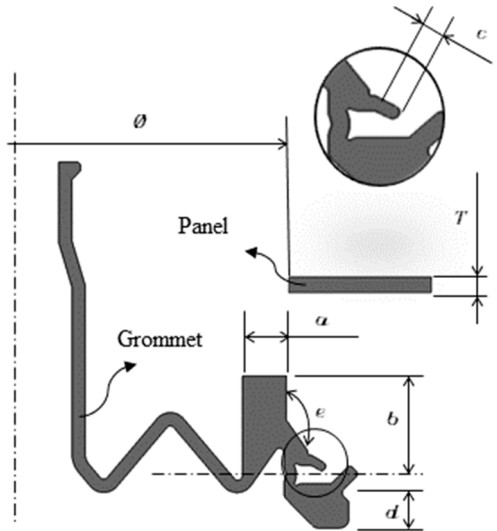

**Figure 5.** Grommet shape parameter settings.

### 3.2. Nonlinear Analysis by Setting Material Constants

Unlike metals, rubber retains its elasticity even under large strain. As rubber has hyper-elastic properties that exhibit nonlinearity between load and strain, it is important to understand its nonlinear properties [15]. We conducted uniaxial and biaxial tensile tests of EPDM rubber to obtain stress–strain rate information. Then, we determined the nonlinear material constants required for finite element analysis. The uniaxial and biaxial tensile tests were performed using an EPDM 50 material to obtain stress–strain data, as shown in Figure 6. We determined the material constants required for nonlinear analysis. The tensile test was performed using the KS standard dumbbell-type three test [13]. The Mullins effect is observed in EPDM materials, such as rubber, in which the initial molecular structure is rearranged upon repeated loads [16–19]. As shown in Figure 6a,b, as strain range gradually increases, if a strain larger than the previously applied strain is received, a certain permanent strain occurs and strain does not become zero, even if stress is zero. In addition, while the gauge distance of a specimen increases in the repeated loading process, the cross-sectional area decreases.

The nonlinear material constants for finite element analysis were obtained through the curve fitting of the stress-strain data obtained in the uniaxial and biaxial tensile tests. The relationship between stress and strain was determined to obtain the final nonlinear material constants considering the change in the cross-sectional area under the repeated loading of rubber, as shown in Figure 7.

The Ogden model was used for nonlinear analysis. An Ogden model is a hyper elastic material model that can be used for predicting the nonlinear stress-strain behavior of materials such as rubber or polymer. Ogden model was introduced by Ogden in 1972, and the strain energy density function for an Ogden material is as follows (3)

$$\mathrm{w} = \sum_{k=1}^{n} \mu_k \left( \frac{\lambda_1^{a_k} + \lambda_2^{a_k} + \lambda_3^{a_k} - 3}{a_k} \right) \tag{3}$$

where w: Strain energy density; $\mu_k$, $a_k$: Ogden constants; and $\lambda$: Stretch ratio; $n = 3$.

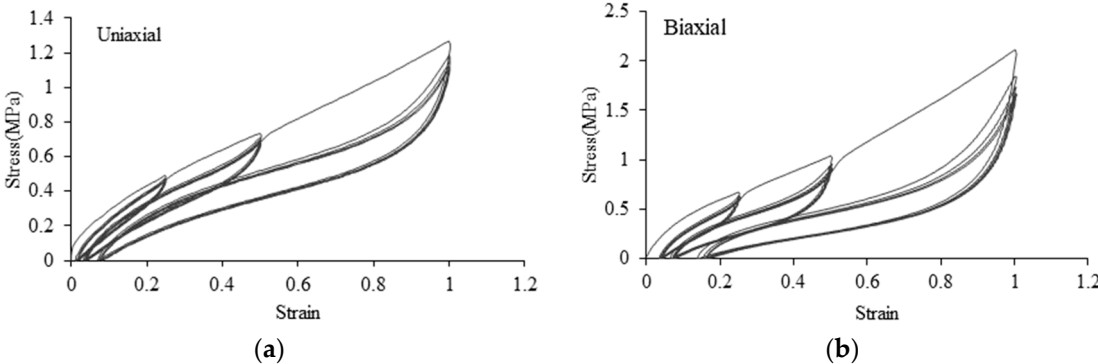

**Figure 6.** Experimental stress–strain curves for compound EPDM. (**a**) Uniaxial tension; (**b**) Biaxial tension.

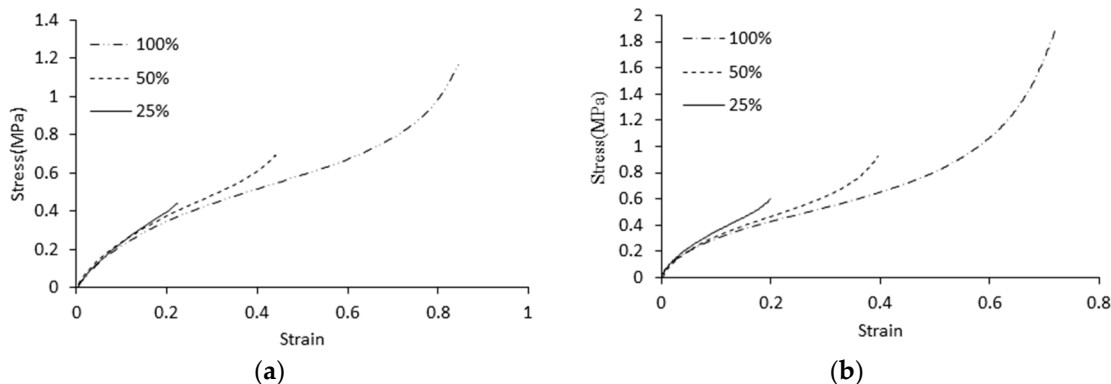

**Figure 7.** Stable stress–strain curves for compound EPDM. (**a**) Uniaxial tension; and (**b**) biaxial tension.

Table 7 shows the coefficients of the Ogden model with a strain range of 100%, which is a material model used to describe nonlinear material constants. The ABAQUS commercial software was used [20–22].

**Table 7.** Ogden model 3rd constant values.

| Material | Ogden Model 3rd Constant Values | | | | | |
|---|---|---|---|---|---|---|
| | μ1 | μ2 | μ3 | α1 | α2 | α3 |
| EPDM50 (100%) | 3.557 | 8.004 | $4.550 \times 10^{-1}$ | $2.000 \times 10^{-2}$ | $4.000 \times 10^{-3}$ | 2.381 |

Figure 8 shows an image of the analysis results. Figure 8a shows the initial state before analysis and (b) shows graphically one of the various analysis results. The maximum value of insertion force and separation force was confirmed through the analysis results. The results for insertion force and separation force were obtained for 32 conditions through the nonlinear analysis, as shown in Table 8. ANOVA was performed for insertion force and separation force according to shape design parameters. Results showed that all variables except d were significant among the variables that affected insertion force, while shape parameters b and e affected separation force.

**Table 8.** Experimental conditions for different dimension values (DOE).

| No. | a [mm] | b [mm] | c [mm] | d [mm] | e [∘] | Insertion Force (N) | Separation Force (N) | No. | a [mm] | b [mm] | c [mm] | d [mm] | e [∘] | Insertion Force (N) | Separation Force (N) |
|-----|--------|--------|--------|--------|-------|---------------------|----------------------|-----|--------|--------|--------|--------|-------|---------------------|----------------------|
| 1 | 4.5 | 11 | 1.5 | 6 | 145 | 62.37 | 69.72 | 17 | 4 | 10 | 2 | 7 | 155 | 64.13 | 70.60 |
| 2 | 4 | 10 | 1 | 7 | 135 | 62.66 | 64.52 | 18 | 4.5 | 11 | 2 | 6 | 145 | 62.95 | 70.21 |
| 3 | 4 | 10 | 2 | 5 | 135 | 61.78 | 64.33 | 19 | 5 | 12 | 2 | 5 | 135 | 62.37 | 70.21 |
| 4 | 5 | 10 | 1 | 7 | 155 | 63.25 | 70.41 | 20 | 4.5 | 11 | 1.5 | 6 | 145 | 62.37 | 69.72 |
| 5 | 4.5 | 11 | 1.5 | 6 | 145 | 62.86 | 70.01 | 21 | 4.5 | 11 | 1 | 6 | 145 | 62.27 | 69.23 |
| 6 | 5 | 11 | 1.5 | 6 | 145 | 62.56 | 70.80 | 22 | 5 | 12 | 2 | 7 | 155 | 63.84 | 78.25 |
| 7 | 5 | 12 | 1 | 7 | 135 | 62.07 | 69.82 | 23 | 4 | 12 | 1 | 7 | 155 | 63.44 | 74.33 |
| 8 | 4.5 | 11 | 1.5 | 6 | 155 | 63.15 | 73.05 | 24 | 4.5 | 11 | 1.5 | 5 | 145 | 62.07 | 69.33 |
| 9 | 5 | 10 | 1 | 5 | 135 | 61.58 | 66.19 | 25 | 4 | 12 | 1 | 5 | 135 | 61.58 | 67.27 |
| 10 | 4 | 10 | 1 | 5 | 155 | 62.27 | 69.03 | 26 | 4.5 | 11 | 1.5 | 6 | 135 | 62.95 | 67.27 |
| 11 | 4.5 | 11 | 1.5 | 6 | 145 | 62.37 | 69.72 | 27 | 4.5 | 11 | 1.5 | 6 | 145 | 62.37 | 69.72 |
| 12 | 4.5 | 11 | 1.5 | 6 | 145 | 62.37 | 69.72 | 28 | 4.5 | 11 | 1.5 | 7 | 145 | 63.35 | 70.31 |
| 13 | 4 | 12 | 2 | 5 | 155 | 62.76 | 74.72 | 29 | 4 | 11 | 1.5 | 6 | 145 | 62.37 | 68.54 |
| 14 | 5 | 10 | 2 | 5 | 155 | 70.31 | 62.76 | 30 | 4 | 12 | 2 | 7 | 135 | 62.95 | 68.15 |
| 15 | 5 | 10 | 2 | 7 | 135 | 62.86 | 66.88 | 31 | 4.5 | 10 | 1.5 | 6 | 145 | 62.27 | 67.37 |
| 16 | 4.5 | 12 | 1.5 | 6 | 145 | 62.37 | 71.78 | 32 | 5 | 12 | 1 | 5 | 155 | 63.15 | 73.15 |

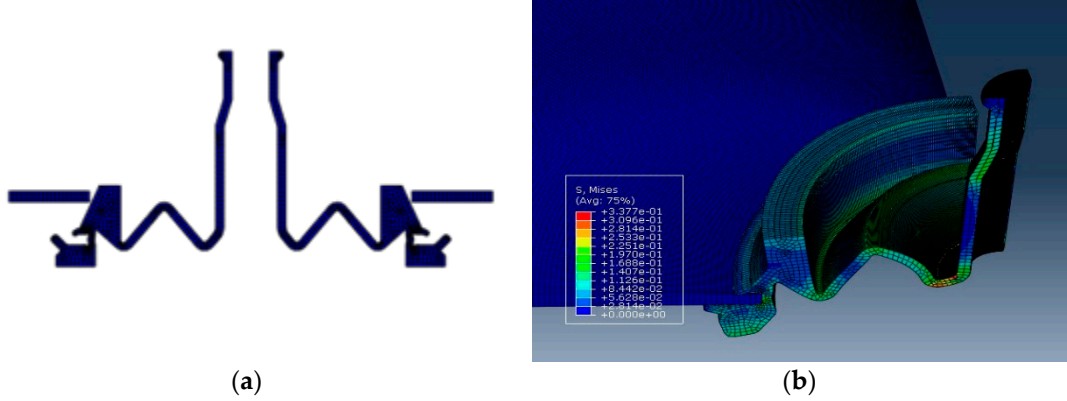

**Figure 8.** Analysis results. (**a**) Sectional view for grommet analysis; and (**b**) analysis result confirmation of stress and maximum stress distribution by the position of grommet.

## 4. Optimization of Shape Parameters

### 4.1. Derivation of Optimal Shape Parameters

We derived the shape parameters for minimizing insertion force and maximizing separation force based on the results obtained from the nonlinear analysis. Large values were obtained for shape parameters a, b, d, and e and a value of 1.313 was obtained for c. Insertion force and separation force were predicted as 62.46 N and 76.98, respectively. Figure 9 is an optimization graph that shows insertion force and separation force according to the values of each shape parameter obtained using the response surface optimization [23].

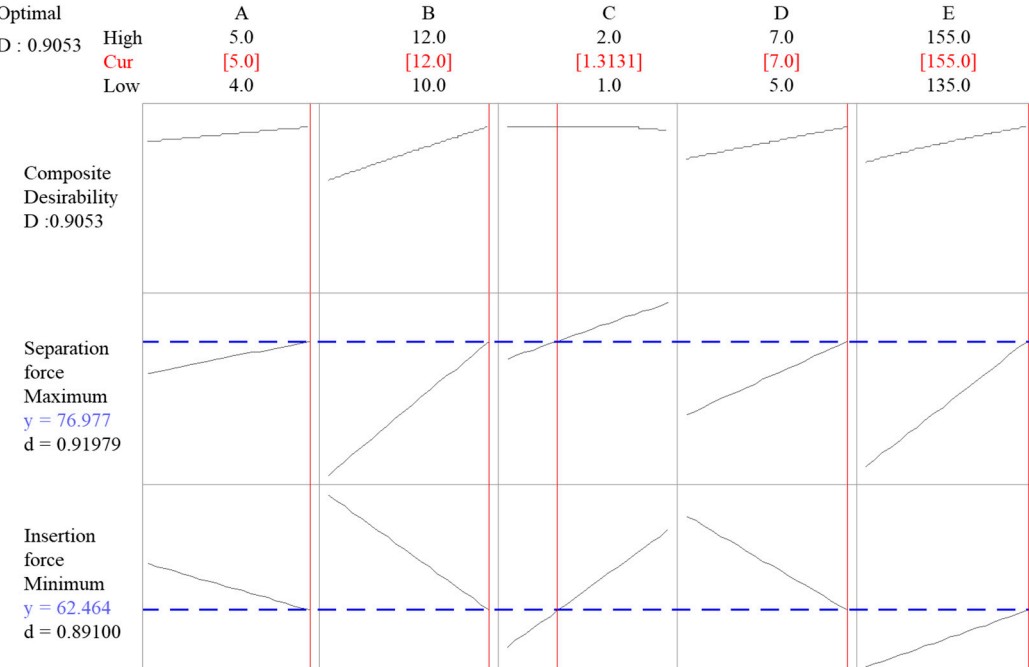

**Figure 9.** Shape parameter optimization results using response optimization tool.

### 4.2. Design Verification

As shown in Table 9, nonlinear analysis was conducted to verify the feasibility of the shape parameters and predicted values were derived using the response optimization tool. To verify the feasibility of the predicted values, nonlinear analysis was conducted by modeling the values of the

derived shape parameters. An insertion force of 62.76 N was derived from the results of the additional nonlinear analysis; the difference from the predicted value (62.46 N) was 0.29 N.

The predicted value of separation force was confirmed to be 77.08 N using the response optimization tool. The value obtained through the additional nonlinear analysis was 76.98 N, and the difference between the predicted and analysis values was 0.10 N. Table 9 shows the predicted values obtained from the response optimization tool and the values obtained via the additional analysis.

**Table 9.** Comparison of predicted and analysis values.

| Classification | Predicted Value (N) | Analysis Value (N) | Difference (N) |
|---|---|---|---|
| Insertion force | 62.46 | 62.76 | 0.29 |
| Separation force | 76.98 | 77.08 | 0.10 |

### 4.3. Verification of Effectiveness

A grommet was fabricated to experimentally test insertion force and separation force using the predicted shape parameters. Based on the process variables set in the previous test, the temperature was set to 170 °C and the time was set to 300 s to produce a grommet. The experiments were conducted as shown in Figure 10. Insertion force was 50.0 N, and separation force was 85.3 N. The predicted and experimental values are different because, when the grommet is fabricated, a protrusion is formed to reduce the friction between the grommet and the mounting part in the insertion part. This causes insertion force to be smaller than the predicted value. Table 10 show the Predicted and measured insertion force and separation force.

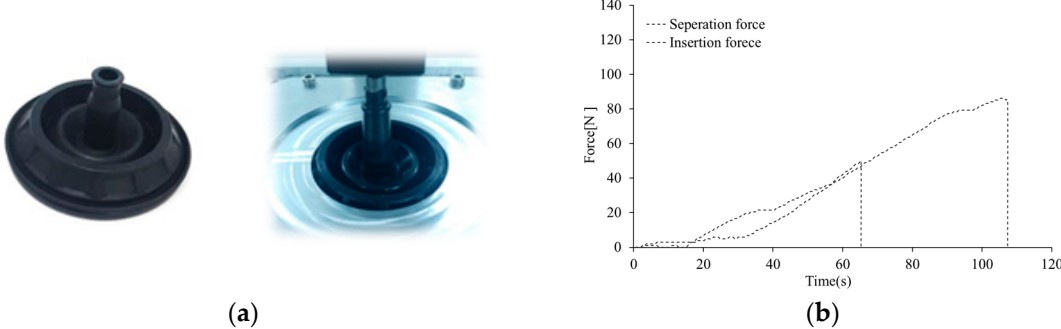

(**a**)                      (**b**)

**Figure 10.** Comparison of simulated and experimental values. (**a**) Insertion force measurement; and (**b**) comparison of insertion and separation forces.

**Table 10.** Predicted and measured insertion force and separation force.

| Classification | Predicted Value (N) | Simulated Value (N) | Experimental Value (N) |
|---|---|---|---|
| Insertion force | 62.5 | 62.8 | 50.0 |
| Separation force | 77.0 | 77.1 | 85.3 |

### 5. Conclusions

This study conducted optimization to improve the mounting performance of a grommet using EPDM materials. The physical properties of the main molding materials were investigated according to process parameters. A grommet was fabricated according to the process parameters of fabrication. Insertion force and separation force were examined through experiments. Nonlinear material constants were determined through uniaxial and biaxial tensile tests. The nonlinear analysis of the grommet was conducted, and a compound design that incorporated the shape parameters for the minimum load of each part was derived. Then, additional nonlinear analysis was performed. This was followed by a comparative analysis of the actual model through experimental evaluation.

1.  The physical properties of EPDM materials were analyzed according to molding parameters. Tensile strength and elongation were measured. Tensile strength increased with temperature and time.

2.  A grommet was fabricated by applying the process parameters that affected the properties of specimens. Experiments were conducted to measure the insertion force and separation force of the fabricated grommet. We confirmed that the maximum load did not change with tensile strength and elongation. Moreover, differences in insertion time occurred owing to differences in elongation.

3.  Uniaxial and biaxial elongation tests of the EPDM materials were conducted to perform the nonlinear analysis of the grommet, and physical property data were derived through the Ogden model. The grommet model was set for each shape parameter and analyzed for various cases. The influence of insertion force and separation force was confirmed through the set shape parameters, and the dimensions for minimizing insertion force and maximizing separation force were derived.

4.  Additional analysis was performed for comparing the results of the optimization and experiments to verify the feasibility of the derived dimensions.

**Author Contributions:** Y.S.K., E.S.H. and E.S.J. conceived and designed the experiments; Y.S.K., E.S.H. and E.S.J. performed the experiments; Y.S.K., E.S.H. and E.S.J. analyzed the data; Y.S.K., E.S.H. and E.S.J. contributed reagents/materials/analysis tools; Y.S.K., E.S.H. and E.S.J. wrote the paper.

**Funding:** This research was financially supported by the Ministry of SMEs and Startups (MSS), Korea, under the "Convergence and Integration R&D(P0005112)" supervised by the Korea Institute for Advancement of Technology (KIAT).

**Conflicts of Interest:** The authors declare no conflict of interest.

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
