# Peer review of "Optimization of Shape Design of Grommet through Analysis of Physical Properties of EPDM Materials"

_applsci, doi:10.3390/app9010133_

Round 1

Reviewer 1 Report

The authors focus on the optimization of the shape design of grommets that are widely uses in industrial applications. Both experimental and modeling tools are covered and several properties have been included. The work is ideally suited for the targeted journal. Below my comments:

L 35 no polarity: better limited

L 42.: plastic; better conventional plastic

L 56: I suggest to add the chemical changes during vulcanization in a figure to increase the readability.

General comment: more details on the molding are needed before the results and discussion. E.g. size; extrusion parameters before the injection, also the temperature of the wall of the mold. Parameters in the table is the injection temperature? The average?

General comment: it would be good to highlight the actual equations after the ANOVA analysis

Please check the positioning of Figure 3.

L 168 the Ogden model needs to better explained.

Figure 8 needs to be covered. Now it is just put there. What are the details?

Reviewer 2 Report

The manuscript illustrates a study on the design optimisation of grommets made of ethylene propylene diene monomer (EPDM) rubber. First, laboratory tests were conducted to determine material properties. Then, design optimisation of shape parameters was performed based on non-linear finite element analysis. Lastly, a prototype grommet was manufactured and its actual performances compared to the theoretical predictions.

The manuscript focuses on a specific technical application with little scientific novelty in terms of new methods and new findings.

Anyway, the following amendments are suggested:

- lines 12-13: remove the instruction text "Featured Application: Authors are encouraged..." or replace it by a meaningful text for the manuscript;

-lines 48-49: replace "an experimental design method" with "a design of experiment (DOE) method" to define the acronym DOE, later used in the captions of Tables 1 and 8;

- Table 1: use SI units (MPa) instead of technical units (kgf/cm^2);

- Fig. 1: use bigger font to make text clearly readable;

- Tables 2 and 3: explain the meaning of all quantities listed;

- Fig. 4: label axis "Force" instead of "Strength" and use SI units (N) instead of technical units (kgf);

- Eq. 1: replace subscripts 1 of the second and third lambdas with 2 and 3, respectively; moreover, define w as the "strain energy density" and specify the value of n = 3;

- Tables 8, 9, 10, and commenting text: use SI units (N) instead of technical units (kgf);

- Fig. 10(b): label axis "Force" instead of "Strength" and use SI units (N) instead of technical units (kgf).

Round 2

Reviewer 2 Report

The manuscript illustrates a study on the design optimisation of grommets made of ethylene propylene diene monomer (EPDM) rubber. First, laboratory tests were conducted to determine material properties. Then, design optimisation of shape parameters was performed based on non-linear finite element analysis. Lastly, a prototype grommet was manufactured and its actual performances compared to the theoretical predictions.

I appreciate the changes introduced by Authors in their revised submission. However, the following typos still need be corrected:

- line 35: "of limited in its chemical structure" --> "of limited polarity in its chemical structure" (?);

- line 49: "design of experimental method(DOE)" --> "design of experiment method (DOE)".

- Table 1: "Mpa" --> "MPa".